# Deletion of Antigen-Presenting Cells in Lipopolysaccharide-Induced Acute Kidney Injury (AKI) Affects the Exacerbation and Repair in AKI

**Jinhai Li, Yuji Nozaki *, Hiroki Akazawa, Kazuya Kishimoto, Koji Kinoshita and Itaru Matsumura**

Department of Hematology and Rheumatology, Faculty of Medicine, Kindai University, Osaka 589-8511, Japan
* Correspondence: yuji0516@med.kindai.ac.jp; Tel.: +81-72-366-0221

**Abstract:** The pathogenesis of acute kidney injury (AKI) is complex and involves various immune and inflammatory responses. Antigen-presenting cells such as macrophages and dendritic cells (DCs) were recently reported to have diverse functions in AKI depending on the pathogenesis and disease phase. Herein, we intraperitoneally administered liposomal clodronate (LC) to lipopoly-saccharide (LPS)-induced AKI model mice in order to deplete antigen-presenting cells (e.g., macrophages and DCs). After the LPS injection, the mice were divided into LC-treated (LPS + LC) and saline-treated groups (LPS), and the immune responses of macrophages and DCs in the acute and recovery phases were evaluated. The LPS + LC-treated group exhibited significantly suppressed renal macrophages and DC infiltration at 18 h and improved survival at 120 h after LPS injection. Via the depletion of macrophages and DC infiltrations, the serum and renal tissue inflammatory cytokines/chemokines were suppressed at 18 h and reversed at 120 h. Tubular kidney injury molecule-1 expression was decreased at 18 h and increased at 120 h. These findings indicate that LC administration suppressed tubular and interstitial injury in the acute phase of AKI and affected delayed tissue repair in the recovery phase. They are important for understanding innate and acquired immune responses in the therapeutic strategy for LPS-induced AKI.

**Keywords:** LPS-induced AKI; antigen-presenting cells; macrophages; dendritic cells; inflammatory cytokine





## 1. Introduction

Acute kidney injury (AKI) is a systemic disease that occurs in >50% of critically ill patients, and it is an independent predictor of mortality in hospitalized patients [1]. Histopathologically, AKI demonstrates acute tubular necrosis (ATN) and infiltrations of inflammatory cells such as neutrophils, lymphocytes, and antigen-presenting cells (APCs) into the tubular stroma; these cells' infiltrations play important roles in the exacerbation of renal failure and tissue repair [2–4]. Investigations of lipopolysaccharide (LPS)-induced AKI have shown the infiltration of APCs into the interstitium, vacuolar degeneration, and apoptosis of tubular cells in renal pathology [5], and the intercellular stimulatory transmission by these cells has been reported to be involved in renal injury [6]. APCs, such as macrophages and DCs, are important regulators of the immune system, as they connect innate immunity and adaptive immunity by inflammatory cytokine pathways [7–9]. Based on previous reports, macrophages and DCs are defined as cell populations with the cell surface markers F4/80 + CD11b+ and F4/80 + CD11c+ [10]. Two types of macrophages with different actions have also been reported [11].

Our present study is the first to evaluate the depletion of macrophages and DCs by liposomal clodronate (LC) in model mice with an LPS-induced AKI in order to clarify the immunological mechanisms underlying the inflammatory cell infiltration in renal pathology that lead to the inhibition of renal injury and repair during the acute and recovery phases

of AKI. Our findings suggest that the determination of the role of APCs in septic kidney injury could be an important contribution to therapeutic strategies.

## 2. Materials and Methods

### 2.1. Experimental Protocol in the Murine Model of LPS-Induced AKI

Figure 1 illustrates the experimental protocol. Male C57BL/6 aged 6–8 weeks (20–25 g body weight) were purchased from The Shizuoka Laboratory Animal Center (Shizuoka, Japan). For the depletion of macrophages and DCs, the mice were injected i.p. with 200 µL/body of LC at 48 h before they received an injection of LPS. The dose of LC was set as the lowest dose that was most effective in depleting the macrophages and DCs in a preliminary experiment. The LC was used in accord with the recommendations from manufacture's protocol.

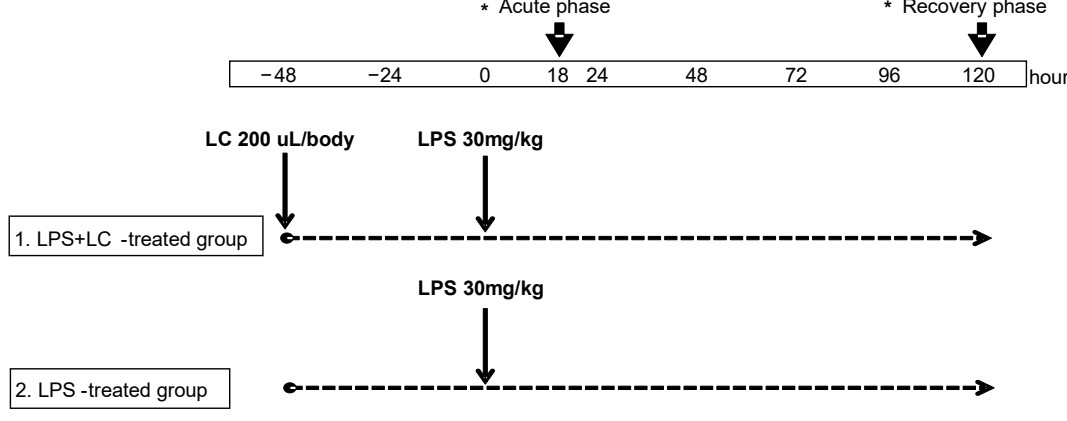

**Figure 1.** Experimental protocol. At 48 h, mice were given either liposomal clodronate (LC) or saline by an intrapertoneal injection. At 0 h, a total of 40 C57BL/6 mice were injected intraperitoneally (i.p.) with 30 mg/kg of lipopolysaccharide (LPS). Mice were culled at 18 ($n = 10$ each) and 120 h ($n = 10$ each). Specimens were collected each hr. Asterisks indicate when specimens were collected.

On day 0, all mice were injected intraperitoneally (i.p.) with 30 mg/kg body weight of LPS (Escherichia coli O111:B4, Sigma-Aldrich, St. Louis, MO, USA). The LPS-injected mice were then randomly divided into two groups (20 mice/group): those to be administered LC (FormuMax Scientific, Palo Alto, CA) (the LPS + LC-treated group) and those to be administered saline alone (the LPS-treated group). The resulting lethality in the LPS + LC- and LPS-treated groups was monitored for 120 h after the LPS injection.

The LPS + LC- and LPS-treated mice were culled at 18 ($n = 10$ each) and 120 h ($n = 10$ each) after the LPS injection with the collection of blood and the collection of kidney tissue. Blood was collected in heparinized tubules for the measurement of blood urea nitrogen (BUN), interferon-gamma (IFN-$\gamma$), tumor necrosis factor (TNF), and interleukin (IL)-18. BUN was measured by an autoanalyzer (Hitachi, Tokyo).

### 2.2. Assessment of Kidney Immunohistochemistry

We used periodic acid–Schiff (PAS) reagent staining to evaluate tubular necrosis in a semiquantitative manner by determining the percentage of cortical tubules in which epithelial necrosis, loss of the brush border, cast formation, and tubular dilation were present. A five-point scale was used in the acute tubular necrosis (ATN) score: 0 points, normal kidney; 1 point, 1–25%; 2 points, 26–50%; 3 points, 51–75%; and 4 points, 76–100% tubular necrosis. Immunohistochemical staining for CD4$^+$, CD8$^+$, and CD68$^+$ cells, and the tubular kidney injury molecule (KIM)-1 was performed on 6-µm-thick periodate

lysine paraformaldehyde-fixed sections [12]. F4/80[+] and CD11c[+] cells were identified in 4-μm-thick formalin-fixed sections [13]. The numbers of these cells were assessed in 10 fields per slide at ×400 magnification, and the results were expressed as cells per high-power field (c/hpf).

The primary antibodies used were rat monoclonal antibody GK1.5 for CD4[+] and 53-6.7 for CD8[+] T cells (BD Biosciences, Franklin Lakes, NJ, USA); F4/80[+] hybridoma culture supernatant (HB198; American Type Culture Collection, Rockville, MD, USA) was used as a pan-macrophage marker; mouse monoclonal anti-CD11c antibody (Abcam, Cambridge, UK) was used for DCs; rat anti-mouse KIM-1 (R&D Systems, Minneapolis, MN) was used as a marker of tubular injury; and mouse CD68[+] (AbD Serotec, Oxford, UK) was used for macrophages. The numbers of CD4[+], CD8[+], CD11c[+], and CD68[+]cells were assessed in 10 glomeruli and interstitia per slide at a magnification of ×400. The numbers of F4/80[+] and KIM-1[+]cells were assessed in the interstitium, and the numbers of tubule-expressing positive cells and cell infiltrates were counted in 10 interstitia per slide at ×400 magnification.

### 2.3. Assessment of Renal mRNA Expression

We performed a real-time polymerase chain reaction (PCR) as described in [13] to measure the intrarenal mRNA expressions of IFN-γ, TNF, IL-10, CCL2/MCP-1, intracellular adhesion molecule-1 (ICAM-1), T-bet, GATA3, and 18SrRNA by using the FastStart DNA Master Sybr Green (Applied Biosystems, Foster City, CA, USA) and the measurement of IL-6, IL-12p40, IL-18, KIM-1, and 18SrRNA by using the TaqMan Gene Expression Assay (Applied Biosystems) on whole kidney tissues. The sequences of primers and the gene database numbers are listed in Tables 1 and 2. The relative amount of mRNA was calculated using the comparative Ct (ΔΔCt) method. All specific amplicons were normalized against 18SrRNA, which was amplified in the same reaction as an internal control using commercial reagents (Applied Biosystems), and the results were expressed as fold differences relative to a saline-treated normal group (*n* = 3).

**Table 1.** Primer sequences (FastStart DNA Master Sybr Green) for analysis of mRNA expression.

| Gene Name | Forward Primer (5′-3′) | Reverse Primer (5′-3′) |
|---|---|---|
| 18SrRNA | GTAACCCGTTGAACCCCATTC | GCCTCACTAAACCATCCAATCG |
| IFN-γ | TGCTGATGGGAGGAGATGTCT | TTTCTTTCAGGGACAGCCTGTT |
| TNF-α | CGATCACCCCGAAGTTCAGTA | GGTGCCTATGTCTCAGCCTCTT |
| IL-10 | GGTTGCCAAGCCTTATCGGA | ACCTGCTCCACTGCCTTGCT |
| CCL2/MCP-1 | AAAAACCTGGATCGGAACCAA | CGGGTCAACTTCACATTCAAAG |
| ICAM-1 | CATCCCAGAGAAGCCTTCCTG | TCAGCCACTGAGTCTCCAAGC |
| T-bet | CCTGGACCCAACTGTCAACT | AACTGTGTTCCCGAGGTGTC |
| GATA3 | AGGGACATCCTGCGCGAACTGT | CATCTTCCGGTTTCGGGTCTGG |

IFN-γ, Interferon-gamma; TNF, Tumor Necrosis Factor; IL, Interleukin; CCL2, Chemokine (C-C Motif) Ligand 2; MCP-1, Monocyte Chemoattractant Chemokine-1; ICAM-1, Intracellular Adhesion Molecule-1.

**Table 2.** Primer sequences (TaqMan Gene Expression Assay) for analysis of mRNA expression.

| Gene Name | TaqMan Gene Expression Assay ID |
|---|---|
| 18SrRNA | 4310893E |
| IL-6 | Mm00446190_m1 |
| IL-12p40 | Mm00434174_m1 |
| IL-18 | Mm00434226_m1 |
| KIM-1 | Mm00506686_m1 |

IL, Interleukin; KIM-1, Kidney Injury Molecule-1.

### 2.4. Cytokine Production

For the measurement of the concentrations of serum IL-18, IFN-γ, and TNF by enzyme-linked immunosorbent assays (ELISAs), we used an anti-mouse IL-18 monoclonal antibody from MBL (Nagoya, Japan), as well asan anti-mouse IFN-γ monoclonal antibody and purified anti-mouse TNF from BD Biosciences as described in [14].

### 2.5. FACS Analysis

For the fluorescence-activated cell sorting (FACS) used to assess the numbers of macrophages and DCs, kidney tissues were taken at the experiment's endpoints, i.e., 18 and 120 h after the LPS injection. Unilateral kidneys were removed from LPS + LC- and LPS-treated mice (18 h; $n$ = 10 each, 120 h; $n$ = 10 each), and renal tissue was properly lysed to remove impurities by the following procedure. Those kidneys were put in collagenase/DNase solution diluted with 2 mL of Roswell Park Memorial Institute medium (RPMI) on ice, and then injected with RPMI. After 25 min of incubation at 37 °C, the kidney samples were mashed with the handle of a syringe and incubated for an additional 25 min. The kidney specimens were then resuspended using a tip until easy pipetting could be performed (mechanical digestion). The volume of the specimen was adjusted to 10 mL in RPMI complete medium, then transferred, spun to wash, and resuspended on a pallet to 10 mL. The pipette tube was kept on ice at 4 °C for 10 min to settle the sediment, and the supernatant was pipetted off and filtered through 40-µm mesh into a new tube.

After red blood cell lysis, all samples (the cell concentration of each specimen was adjusted to $2 \times 10^6$ cells/mL) were washed in 1% bovine serum albumin (BSA) in phosphate-buffered saline (PBS), blocked by Mouse Fc BlockTM Reagents (BD Biosciences), and stained with either a cocktail containing PE-Vio770®-conjugated anti-F4/80, APC-Vio®770-conjugated anti-CD11c (Miltenyi Biotec, Bergisch-Gladbach, Germany) and BV421-conjugated anti-CD11b (BD Biosciences), or a cocktail of the corresponding isotype controls. The number of mononuclear cells extracted was counted and gated in the FACS analysis. Samples were run on a BD FACS Canto II Flow Cytometer. A saline-treated normal group ($n$ = 6) served as the control, and a liposome + PBS-treated group ($n$ = 4) served as the sham animals for this experiment.

### 2.6. Statistical Analysis

The results are expressed as the mean ± SEM (standard error of the mean). Groups were compared by the unpaired t-test or by an analysis of variance (ANOVA) when more than two groups were compared. Probability values <0.05 were accepted as significant. The survival time was estimated by the Kaplan–Meier method. The log-rank test was used to compare survival times between groups. The data were analyzed using GraphPad Prism software ver. 6.0 (GraphPad Software, La Jolla, CA, USA).

## 3. Results

### 3.1. The LC Treatment Depleted Renal Macrophages and Dendritic Subset in the Mice

Figure 2 shows the results of the FACS analysis, i.e., the percentages of renal macrophages and DCs in the LPS- and LPS + LC-treated groups at 18 and 120 h after LPS injection. The FACS analysis defined the cells gating with F4/80+ as CD11b+ cells and CD11c+cells as macrophages and DCs in the kidney. The administration of LC decreased the percentage of renal macrophages at 18 h after the LPS injection (LPS vs. LPS+LC groups, 5.0% vs. 3.2%, respectively) and DCs (3.1% vs. 1.4%, respectively). Conversely, the percentages of renal macrophages in the LPS + LC-treated mice (2.7% vs. 1.4%) and those of DCs (1.0% vs. 0.4%) were both increased compared to those in the LPS-alone group at 120 h after the LPS injection. Renal macrophages and DCs in the liposome + PBC-treated sham group were 8.4% and 6.7%, respectively, and did not differ significantly from the normal group treated with the saline.

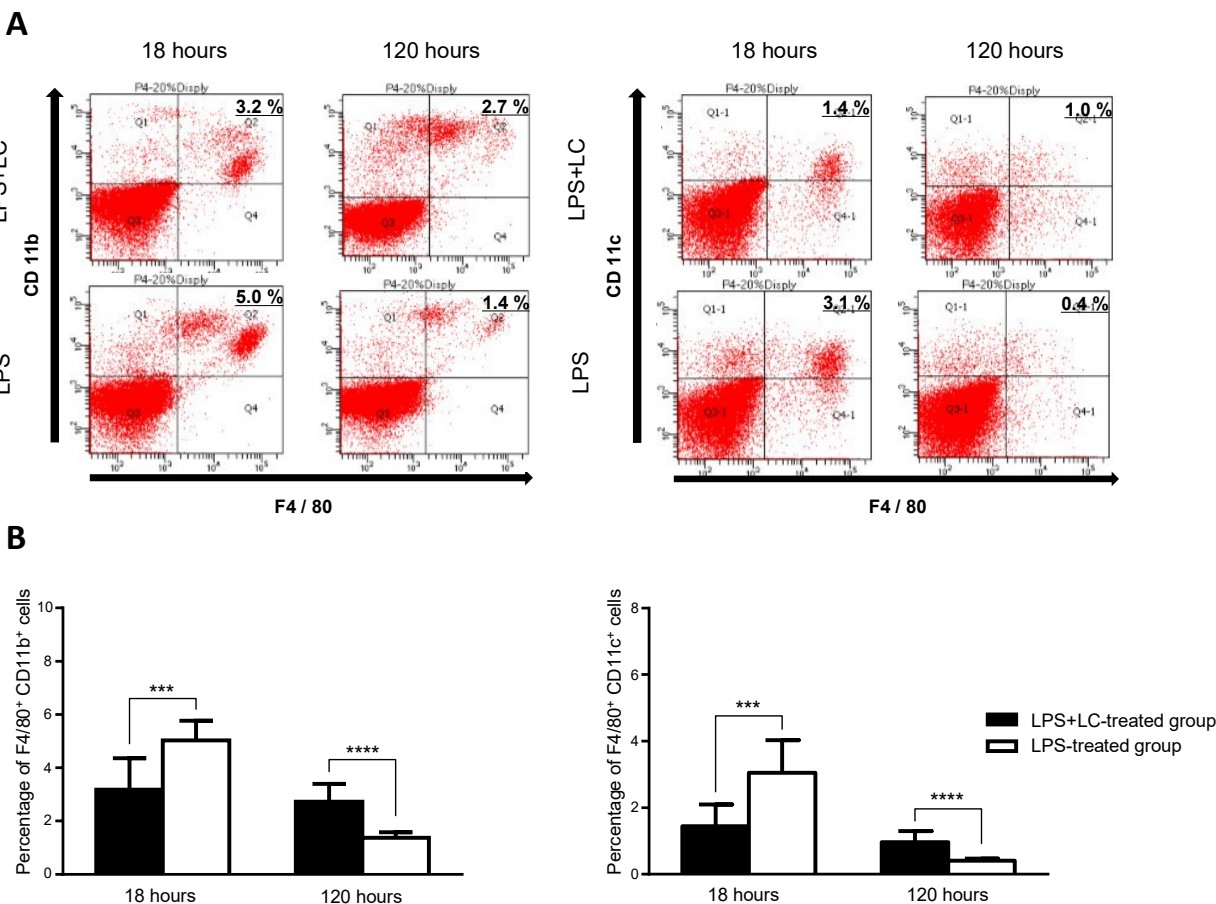

**Figure 2.** Liposomal clodronate depleted CD11b$^+$ and CD11c$^+$ cells gating with F4/80$^+$cells. (**A**): At 18 and 120 h after the LPS injection, cell suspensions were obtained from the kidney of LPS + LC-treated and LPS-treated mice ($n$ = 10 each). Representative data from the FACS analysis are shown. (**B**): The percentages of CD11b$^+$ F4/80$^+$ and CD11c$^+$F4/80$^+$ cells. The data are mean ± SEM. *** $p < 0.005$, **** $p < 0.001$, LPS + LC vs. LPS-treated group.

### 3.2. Macrophage and DC Depletion Improved the Survival Rate by Inhibiting the Renal Injury after LPS-Induced AKI

To determine the mortality in LPS-induced AKI, we evaluated the survival rate of the LPS- and LPS + LC-treated mice for 120 h after LPS injection. The results demonstrated a significantly higher survival rate in the LPS + LC-treated mice compared to those treated with LPS alone (50.0% vs. 16.1%, $p < 0.01$) (Figure 3A). The increase in BUN levels by LPS at 18 h was significantly reduced in the LPS + LC-treated group compared to the LPS-treated group (81.7 ± 1.9 vs. 89.6 ± 2.5 mg/dL, $p < 0.05$). Conversely, BUN levels at 120 h were significantly increased in the LPS + LC-treated group compared to the LPS-treated group (37.2 ± 1.2 vs. 30.7 ± 1.0 mg/dL, $p < 0.005$). The effect of the LC on renal function was also examined, and LC mice injected with saline instead of LPS were set up as the sham group ($n$ = 4 each). We observed that there was no significant difference in BUN levels in the sham group compared to normal mice. (Figure 3B).

The degree of renal injury was evaluated based on the ATN score. At 18 h after LPS injection, kidneys from the mice showed some tubular dilation in the cortical tubules plus loss of the tubular brush border, but the LPS + LC-treated kidneys had slightly better-preserved brush borders and less tubular epithelial cells compared to the LPS-treated group (Figure 3C). In the LPS + LC-treated group, the ATN score was decreased at 18 h but increased at 120 h compared to the LPS-treated group (Figure 3D).

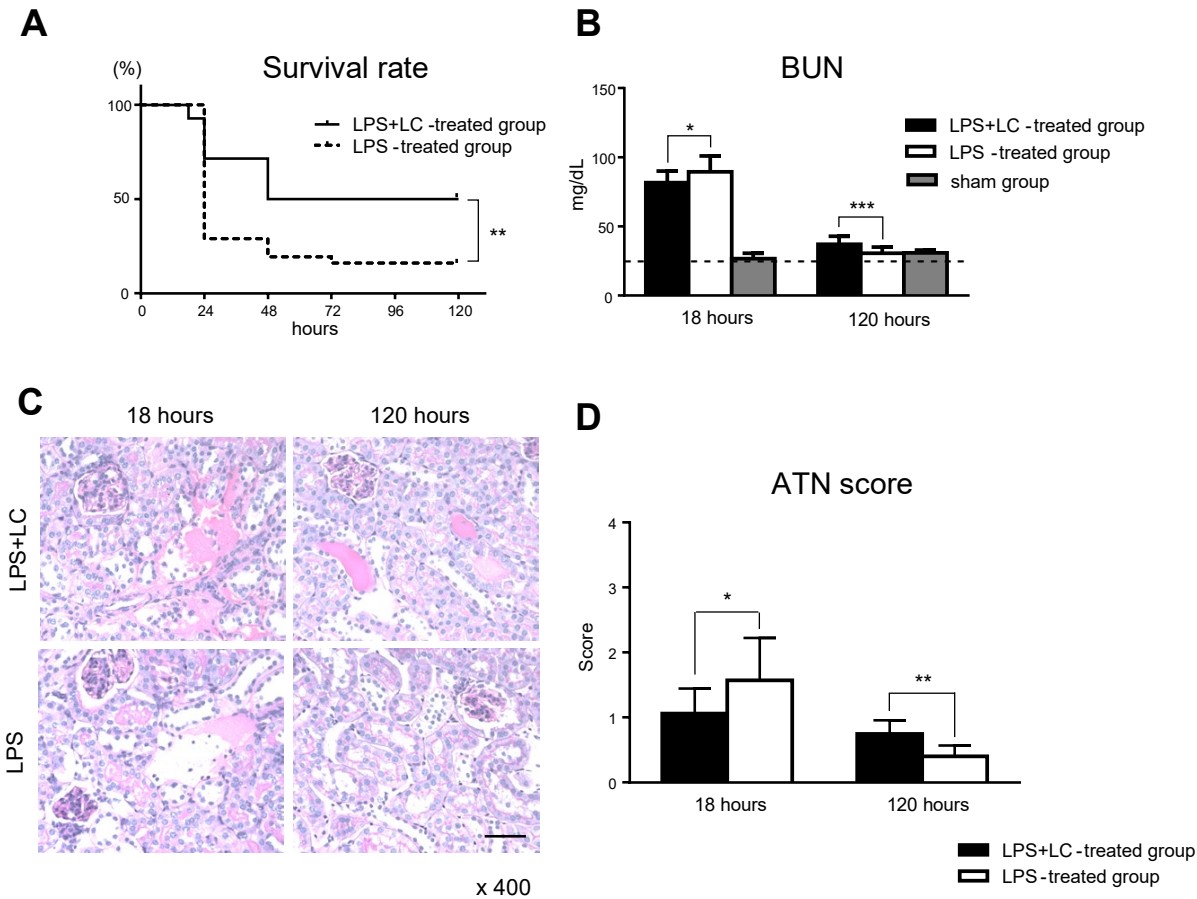

**Figure 3.** Effect of LC on the survival and the BUN level in the development of AKI after LPS injection. (**A**): Survival of the LPS + LC-treated mice and LPS-treated mice subjected to AKI by LPS injection. Mice were evaluated until 120 h after the LPS injection. (**B**): Renal function of the LPS + LC-treated mice and LPS-treated mice at 18 and 120 h after LPS injection, assessed by BUN levels. LC-treated mice that received saline instead of LPS were designated as the sham group, and BUN levels were also measured. (**C**): The grade of renal injury was evaluated by periodic acid–Schiff staining. Kidneys were stained with periodic acid–Schiff reagent to evaluate tubular necrosis in a semiquantitative manner by determining the percentage of cortical tubules in which epithelial necrosis, loss of the brush border, cast formation, and tubular dilation were observed; ×400. (**D**): The grade of renal injury in the LPS + LC- and LPS-treated groups at 18 and 120 h after LPS injection was evaluated by the acute tubular necrosis (ATN) score. The data are mean ± SEM. * $p < 0.05$, ** $p < 0.01$, *** $p < 0.005$, LPS + LC- vs. LPS-treated group. Scale bar, 100 μm. AKI: acute kidney injury, BUN: blood urea nitrogen.

### 3.3. LC Treatment Affected Renal Injury, the Infiltration of Inflammatory Cells, and Tubular Damage in the Mouse Kidney after LPS Injection

The immunohistochemical staining of kidneys revealed the infiltration of F4/80+ cells as a pan-macrophage marker after LPS injection (Figure 4A,B). At 18 h, the number of F4/80+ cells in the interstitium were reduced in the LPS + LC-treated group compared to the LPS-treated group. However, at 120 h, the LPS + LC-treated group showed a significant increase in F4/80+ cells compared to the LPS-treated group. Regarding CD4+ and CD8+ T-cells, there were no significant differences between the two groups at 18 or 120 h (Figure 5A). The number of infiltrating CD68+ cells was increased in the glomeruli and interstitium in the LPS-treated group versus the LPS + LC-treated group at 18 h.

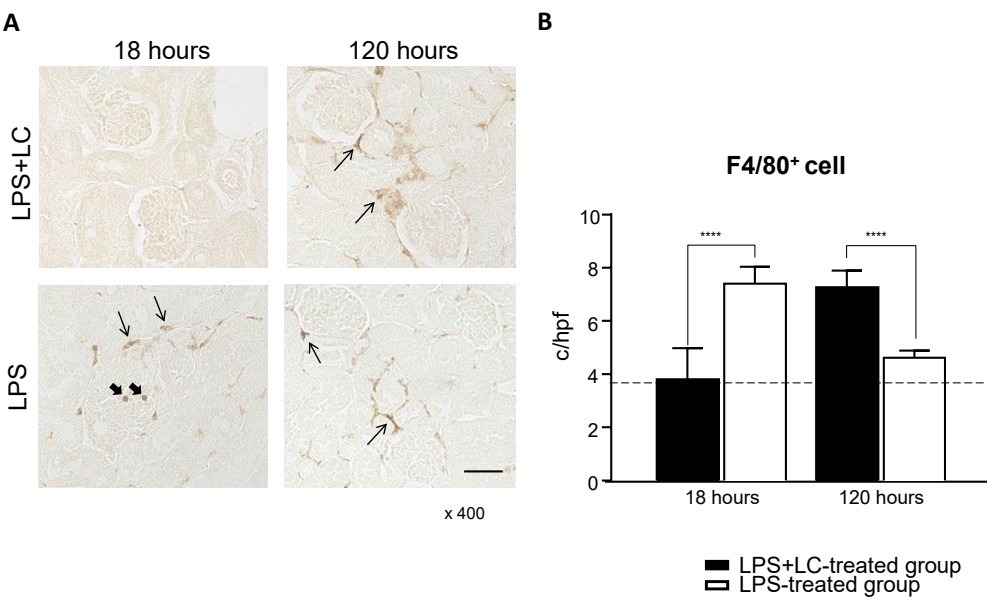

**Figure 4.** Effect of LC on the accumulation of inflammation F4/80+cells in the interstitium after LPS injection. (**A**): Representative photograph of F4/80+ cells in the kidney from an LPS + LC-treated mouse and an LPS-treated mouse after LPS injection. The infiltrations of F4/80+ cells in the interstitium in the LPS + LC and LPS-treated groups after LPS injection. In the periglomerular area, F4/80+ cells infiltrated and are presented as brown dots. (**B**): The numbers of F4/80+ cell infiltrates in the renal interstitium are shown at 18 and 120 h. The data are mean ± SEM. **** $p < 0.001$, LPS + LC vs. LPS-treated group. c/hpf: cells per high-power field.

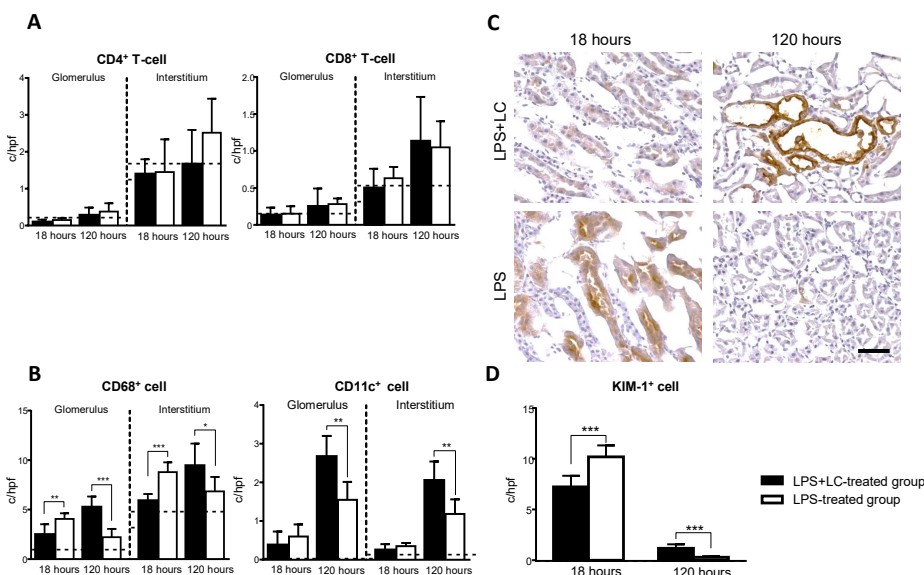

**Figure 5.** Effect of LC on T-cell and DC accumulation and KIM-1 expression in the glomerulus and interstitium after LPS injection. (**A,B**): At 18 and 120 h post-LPS injection, the numbers of CD4+ and CD8+ T cells and the numbers of CD68+ and CD11c+cells as macrophages and DCs (c/hpf) in the glomeruli and interstitia in the LPS + LC- and LPS-treated groups. (**C**): KIM-1 expression was detected in many tubules in the injured interstitium after LPS injection, localized to the apical side of the epithelium with some diffuse cytoplasmic staining (brown) (×400 magnification). (**D**): KIM-1+ cells were also counted and the numbers of positive cells in the LPS + LC- and LPS-treated groups were compared. Dotted lines: Mean values from the saline-injected group without LPS. The data are mean ± SEM. * $p < 0.05$, ** $p < 0.01$, *** $p < 0.005$, LPS + LC vs. LPS-treated group. KIM-1: kidney injury molecule-1.

Conversely, in the LPS + LC-treated group, the numbers of CD68[+] and CD11c[+] cells were significantly increased in the glomeruli and interstitium at 120 h compared to the LPS-treated group (Figure 5B). At 18 h, the number of KIM-1[+] cells as a marker of tubular damage were significantly lower in the interstitium of the LPS + LC-treated mice compared to the LPS-treated group. Conversely, at 120 h, these cells were significantly increased in the LPS + LC-treated group compared to the LPS-treated group (Figure 5C).

*3.4. Serum Levels of Inflammatory Cytokines and the Intrarenal mRNA Expression of an Inflammatory Mediator after the LPS Injection*

At 18 h post-LPS injection, the serum IL-18, IFN-γ, and TNF levels were significantly decreased in the LPS + LC-treated mice compared to those treated with only LPS. At 120 h post-LPS injection, however, the serum IL-18 and TNF levels were significantly increased in the LPS + LC-treated group, and the serum IFN-γ level tended to be increased compared to the LPS-treated group (Figure 6). Table 3 provides the renal mRNA expression levels of cytokines, chemokines, KIM-1, Th cell subset transcription factors, and leukocyte adhesion molecules revealed by the real-time PCR. The mRNA expressions of cytokines (IFN-γ, TNF, IL-6, and IL-18), chemokines (CCL2/MCP-1) and KIM-1, the Th cell subset transcription factor T-bet, and the leukocyte adhesion molecule ICAM-1 were all significantly decreased at 18 h in the LPS + LC-treated group compared to the LPS-treated group. Conversely, at 120 h, the mRNA expressions of IL-6, MCP-1, and KIM-1 were increased in the LPS + LC-treated mice compared to LPS-treated mice. The renal expressions of IL-10, IL-12p40, and GATA3 mRNA were not significantly different between the groups at 18 or 120 h.

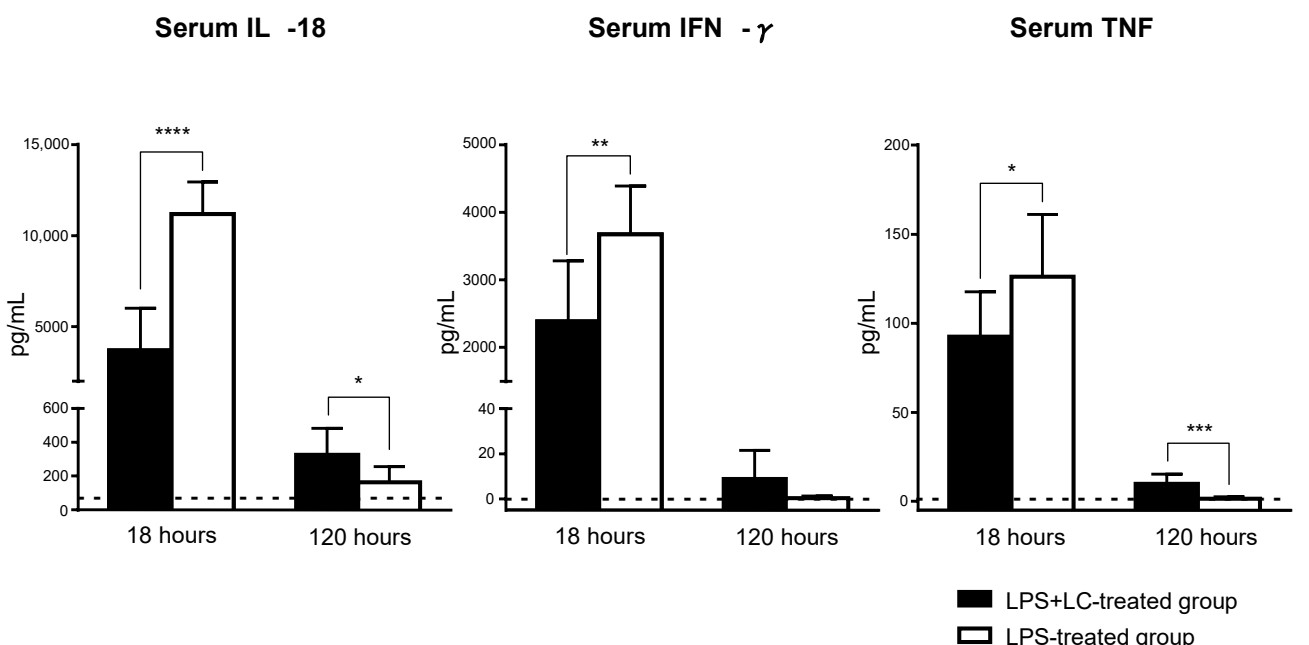

**Figure 6.** Effect of LC on serum proinflammatory cytokines after LPS injection. Serum IL-18, IFN-γ, and TNF were measured as biomarkers in AKI. Dotted lines: Mean values from saline-injected group without LPS. The data are mean ± SEM. * *p* < 0.05, ** *p* < 0.01, *** *p* < 0.005, **** *p* < 0.001, LPS- vs. LPS + LC-treated group.

**Table 3.** Renal mRNA expression of cytokines, chemokines, KIM-1, Th cell subset transcription factors, and leukocyte adhesion molecules by real-time PCR.

|  | 18 h | 120 h |
|---|---|---|
|  | **LPS + LC vs. LPS Treated Group** | |
| Cytokines | | |
| IFN-γ | 14.7 ± 1.6 vs. 23.6 ± 3.0 * | 4.0 ± 0.6 vs. 3.0 ± 0.5 |
| TNF | 24.9 ± 2.6 vs. 31.5 ± 1.2 * | 12.2 ± 3.3 vs. 8.0 ± 1.7 |
| IL-6 | 1314.0 ± 190.7 vs. 3900.0 ± 580.8 **** | 42.6 ± 9.8 vs. 17.4 ± 3.3 * |
| IL-10 | 56.2 ± 15.3 vs. 36.3 ± 11.1 | 5.8 ± 1.9 vs. 3.5 ± 0.5 |
| IL-12p40 | 0.6 ± 0.1 vs. 0.6 ± 0.1 | 0.9 ± 0.2 vs. 1.0 ± 0.1 |
| IL-18 | 5.9 ± 0.8 vs. 10.8 ± 0.9 ** | 1.8 ± 0.4 vs. 1.3 ± 0.1 |
| Chemokines | | |
| CCL2/MCP-1 | 75.0 ± 10.8 vs. 153.4 ± 18.5 ** | 13.4 ± 2.2 vs. 6.5 ± 1.5 * |
| KIM-1 | 89.8 ± 16.0 vs. 258.7 ± 45.1 ** | 26.6 ± 16.8 vs. 2.7 ± 1.5 ** |
| Th cell subset transcription factors | | |
| T-bet | 1.7 ± 0.3 vs. 3.5 ± 0.6 * | 3.7 ± 1.0 vs. 5.2 ± 0.6 |
| GATA3 | 0.9 ± 0.1 vs. 0.9 ± 0.1 | 0.9 ± 0.1 vs. 1.1 ± 0.1 |
| Leukocyte adhesion molecule | | |
| ICAM-1 | 61.3 ± 6.7 vs. 101.5 ± 16.7 * | 1.9 ± 0.3 vs. 1.9 ± 0.3 |

Each group of renal mRNA of cytokines (IFN-γ, TNF, IL-6, IL-10, IL-12p40 and IL-18), chemokines (CCL2/MCP-1), Th cell subset transcription factors (T-bet and GATA3), and leukocyte adhesion molecule ICAM-1 were measured at 18 and 120 h after LPS injection by real-time PCR. All specific amplicons were normalized against 18SrRNA, which was amplified in the same reaction as control mice treated with saline. The data are the mean fold-increase ± SEM (* $p < 0.05$, ** $p < 0.01$ and **** $p < 0.001$; LPS + LC vs. LPS-treated group ($n = 8$ per group)). IFN-γ, Interferon-gamma; TNF, Tumor Necrosis Factor; IL, Interleukin; CCL2, Chemokine (C-C Motif) Ligand 2; MCP-1, Monocyte Chemoattractant Chemokine-1; KIM-1, Kidney Injury Molecule-1; ICAM-1, Intracellular Adhesion Molecule-1.

## 4. Discussion

Macrophages are a major source of inflammatory cytokines, such as IFN-γ, TNF, IL-6, and IL-18, and chemokines, such as MCP-1, all of which contribute to kidney injury and the promotion of kidney repair and fibrosis after injury [15–18]. There are reports about the activation of kidney-resident DCs during the early stages of ischemia–reperfusion injury (IRI) [19,20] and the involvement of DCs in kidney injury via the secretion of TNF-α and other inflammatory cytokines [21]. The functional importance of macrophages and DCs in AKI has been shown to have an important role in IRI-induced AKI, in both the acute and recovery phases [21,22].

In addition, MCP-1/CCL2, which are important chemokines that regulate the migration and infiltration of monocytes and macrophages, have been found to decrease renal mRNA expressions in response to a decrease in inflammatory cytokines by macrophages [5]. These reports are compatible with our present observation that LC treatment may have inhibited the progression of AKI by depleting APCs via the suppression of inflammatory cytokines, chemokines, and cell adhesion molecules in the acute phase. On the other hand, our analyses demonstrated that renal dysfunction remained in the LPS + LC-treated mice, even after the normalization of the BUN levels in the LPS-treated group at 120 h after the injection of LPS. At 120 h, the LPS + LC-treated mice also showed increased serum cytokine levels (IFN-γ, TNF, and IL-18), renal MCP-1 and inflammatory cytokine mRNA expressions, and histological renal tubular injury compared to the LPS-treated group. We consider the possibility that the depletion of macrophages in the early phase of AKI may have been responsible for the decrease in M2 macrophages in the late phases, resulting in delayed tissue repair.

Our present findings indicate that (i) DCs and macrophages are closely related to immunological responses during the acute and recovery phases in LPS-induced AKI, and (ii) macrophage differentiation may influence tissue injury and tissue repair. However, it is difficult to strictly separate the cell surface markers of macrophages (F4/80+ CD11b+) and

those of DCs (F4/80$^+$ CD11c$^+$), since they share some common populations. In addition, this experiment focused only on macrophages and dendritic cells, and did not discriminate APCs or lymphocytes. These are issues to be addressed in further investigations. The present experimental protocol aimed to deplete macrophages and DCs by the administration of LC to LPS-induced AKI mice, but future studies are needed to evaluate cell-specific functions. For this reason, we believe it is important to evaluate the creation of AKI models and macrophage depletion using other alternative pathways, and we were also conducting parallel experiments using IRI mice as a model for AKI, but many mice died due to invasive laparotomy or arterial occlusion. We also thought that the same practitioner could see variations in technique from day to day, and in fact, BUN levels were not constant. We considered IRI mice unsuitable for evaluation as an AKI model. We have not been able to conduct experiments using reagents such as anti-CSF-1R antibodies as a means of depleting macrophages, and would like to consider additional experiments in the future. The removal of macrophages and dendritic cells suppresses the immune response and delays the progression of AKI, but since it also suppresses M2 macrophages, it may contribute to tissue retardation. The therapeutic efficacy and adverse effects must be considered for each pathological condition and stage with regard to clinical application, which is also an issue for future research.

## 5. Conclusions

Our results demonstrated that, in LPS-induced AKI, macrophages and DCs have different roles in tissue damage and the associated repair via the production of inflammatory cytokines and chemokines.

**Author Contributions:** Conceptualization, Y.N.; Data curation, J.L.; Formal analysis, J.L.; Investigation, J.L.; Methodology, Y.N.; Project administration, Y.N.; Supervision, Y.N., K.K. (Kazuya Kishimoto) and I.M.; Validation, J.L., Y.N., K.K. (Kazuya Kishimoto) and K.K. (Koji Kinoshita); Visualization, J.L.; Writing—original draft, J.L.; Writing—review & editing, J.L., Y.N. and H.A. All authors have read and agreed to the published version of the manuscript.

**Funding:** This research received no external funding.

**Institutional Review Board Statement:** The study was conducted according to the guidelines of the Declaration of Helsinki, and the animal protocols were approved by and performed in accordance with the Kindai University Animal Care Guidelines (KAME 30-011).

**Informed Consent Statement:** Not applicable.

**Data Availability Statement:** Not applicable.

**Acknowledgments:** We thank Shinji Kurashimo and Nobuyuki Mizuguchi for their expert technical assistance.

**Conflicts of Interest:** The authors declare no conflict of interest.

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
