# Peer review of "Deletion of Antigen-Presenting Cells in Lipopolysaccharide-Induced Acute Kidney Injury (AKI) Affects the Exacerbation and Repair in AKI"

_cimb, doi:10.3390/cimb44110383_

Round 1
Reviewer 1 Report
Comments:
In this manuscript by Li and colleagues, the authors evaluate the depletion of macrophages and DCs by liposomal clodronate (LC) in model mice with an LPS-induced AKI and clarify the immunological mechanisms underlying the inflammatory cell infiltration in renal pathology that leads to the inhibition of renal injury and repair during the acute and recovery phases of AKI. They investigate the affection of the exacerbation and repair in AKI by deleting of antigen-presenting cells in lipopolysaccharide-induced acute kidney injury (AKI).
The authors have performed a investigation in this manuscript in determining the different roles of macrophages and DCs in LPS-induced AKI. The study is well conducted, and the conclusions are well supported by the data and this manuscript can be accepted for publication with the following minor comments addressed
1.In line 134, the number representing the power of ten is not superscripted. 2× 106 could be instead of 2× 106 cells.
2.****p<0.001 should be added in the legend of Figure6(line 254)
3.In Materials and Methods, it is better to illustrate all primers sequences used in Table3
Author Response
Response1, 2: We have corrected the points you indicated.
Response3: For PCR of IL-6, IL-12, IL-18 and KIM-1 using TaqMan Gene Expression Assays, we are sorry, but we do not know the primer sequence and it is difficult to show.
Reviewer 2 Report
Using a rigorous and well-designed protocol, authors demonstrate, with a detailed description of the results, the bi-modal response of macrophages after depletion with a single dose of liposomal clodronate (LC) in lipopolysaccharide (LPS) -induced AKI model mice.
The tables, graphs and figures are appropriately presented.
The main weakness of the manuscript is the lack of comments in the discussion section about the differences and potential advantages of this specific model (LC+LPS-AKI) compared with the use of LC in other AKI models (unilateral ureteral obstruction, ischemia-reperfusion injury, etc.) or the adoption of alternative pathways as anti-CSF-1R antibodies to deplete macrophages.
Another question to the authors is how to apply the findings from this AKI model into clinical practice.
Minor comments:
APCs not previously defined in abstract line 14
Authors should revise Graphic B in Figure 3. They include a LC group not previously described.
Author Response
Point 1: The main weakness of the manuscript is the lack of comments in the discussion section about the differences and potential advantages of this specific model (LC+LPS-AKI) compared with the use of LC in other AKI models (unilateral ureteral obstruction, ischemia-reperfusion injury, etc.) or the adoption of alternative pathways as anti-CSF-1R antibodies to deplete macrophages.
Response 1: In parallel with this experiment, we had also conducted experiments using ischemia-reperfusion injury (IRI) mice as a model of AKI, but many mice died due to invasive laparotomy or arterial occlusion. We also considered that even the same practitioner could see variations in technique from day to day, and in fact, BUN values were not constant in each IRI mouse. Based on the above experience, we think this is inefficient compared to using lipopolysaccharide.
We have not been able to experiment or study alternative pathways to deplete macrophages, such as anti-CSF-1R antibody, due to time constraints. We would like to consider them in future experiments.
I have added the above comments to the discussion section.
Point 2: Another question to the authors is how to apply the findings from this AKI model into clinical practice.
Response 2: It is very important to point out whether the clinical application is feasible or not. In the acute phase, depletion of macrophages and dendritic cells suppresses the immune response and delays AKI progression, but it may also suppress M2 macrophages, which may abet to delayed tissue repair. Therefore, future work needs to examine in what disease, or even if the same disease, at what stage macrophage depletion is necessary, in terms of therapeutic efficacy and detrimental effects.
Minor comments
Point 1: APCs not previously defined in abstract line 14.
Response 1: We have corrected APCs in the Abstract to antigen-presenting cells as pointed out.
Point 2: Authors should revise Graphic B in Figure 3. They include a LC group not previously described.
Response 2: We apologize for not explaining the details of Fig. 3B properly. We added the following in the text of Results 3-2: " The effect of LC on renal function was also examined, and LC mice injected with saline instead of LPS were set up as Sham group (n = 4 each). We observed that there was no significant difference in BUN levels in Sham group compared to normal mice. " Graphic B and legends in Figure 3 were also corrected and added, respectively.